# Gastroschisis: A State-of-the-Art Review

**DOI:** 10.3390/children7120302

**Published:** 2020-12-17

**Authors:** Vishwanath Bhat, Matthew Moront, Vineet Bhandari

**Affiliations:** 1Division of Neonatology, Department of Pediatrics, The Children’s Regional Hospital at Cooper, Cooper Medical School of Rowan University, One Cooper Plaza, Camden, NJ 08103, USA; bhat-vishwanath@cooperhealth.edu; 2Division of Pediatric Surgery, Department of Pediatrics, The Children’s Regional Hospital at Cooper, Cooper Medical School of Rowan University, One Cooper Plaza, Camden, NJ 08103, USA; moront-matthew@cooperhealth.edu

**Keywords:** abdominal defect, infant, newborn, congenital anomaly, nutrition

## Abstract

Gastroschisis, the most common type of abdominal wall defect, has seen a steady increase in its prevalence over the past several decades. It is identified, both prenatally and postnatally, by the location of the defect, most often to the right of a normally-inserted umbilical cord. It disproportionately affects young mothers, and appears to be associated with environmental factors. However, the contribution of genetic factors to the overall risk remains unknown. While approximately 10% of infants with gastroschisis have intestinal atresia, extraintestinal anomalies are rare. Prenatal ultrasound scans are useful for early diagnosis and identification of features that predict a high likelihood of associated bowel atresia. The timing and mode of delivery for mothers with fetuses with gastroschisis have been somewhat controversial, but there is no convincing evidence to support routine preterm delivery or elective cesarean section in the absence of obstetric indications. Postnatal surgical management is dictated by the condition of the bowel and the abdominal domain. The surgical options include either primary reduction and closure or staged reduction with placement of a silo followed by delayed closure. The overall prognosis for infants with gastroschisis, in terms of both survival as well as long-term outcomes, is excellent. However, the management and outcomes of a subset of infants with complex gastroschisis, especially those who develop short bowel syndrome (SBS), remains challenging. Future research should be directed towards identification of epidemiological factors contributing to its rising incidence, improvement in the management of SBS, and obstetric/fetal interventions to minimize intestinal damage.

## 1. Introduction

In recent years, the subject of gastroschisis has received considerable attention due to: (1) the controversy surrounding its pathogenesis in light of our understanding of its developmental biology; (2) a temporal increase in incidence of gastroschisis for reasons that are not entirely clear; (3) improvements in our ability to diagnose the condition early in pregnancy and identify “high-risk” fetuses; (4) improvements in operative techniques; and (5) advances in parenteral and enteral nutritional support during the postoperative period.

The first case of an abdominal wall defect was reported by Conrad Wolffhart (Lycosthenes), an Alsatian humanist and theologist, who described an infant, born in 1547, with “a large extrusion of intestines from abdomen and chest, feet by the head, and a tall, pointed skull” [1]. The term “gastroschisis” (from ancient Greek, gastro = stomach, and schisi = split), first coined by Taruffi in 1894, is actually a misnomer, since it is the anterior abdominal wall, not the stomach, that is split. In 1953, Moore and Stokes first described the features differentiating gastroschisis from omphalocele [2].

## 2. Definition

The International Clearinghouse for Birth Defects Surveillance and Research defines gastroschisis (Figure 1) as “a congenital malformation characterized by visceral herniation usually through a right side abdominal wall defect to an intact umbilical cord and not covered by a membrane” [3]. Approximately 10% of infants with gastroschisis have intestinal stenosis or atresia [4] resulting from vascular insufficiency due to a volvulus or compression of vascular pedicle by a narrowing abdominal ring [5].

## 3. Pathogenesis

Several embryologic hypotheses have been proposed to elucidate the pathogenesis of gastroschisis including a failure of differentiation of the embryonic mesenchyme due to a teratogenic exposure [6], rupture of the amniotic membrane at the base of the umbilical cord [7], abnormal involution of the right umbilical vein, leading to impaired viability of surrounding mesenchyme [8], interruption of the omphalomesenteric artery leading to localized necrosis of the abdominal wall at the base of the cord [9], abnormal folding of the embryo leading to a ventral body wall defect [10], failure of the yolk sac and related vitelline structures to be incorporated into the umbilical stalk, leading to a perforation in the abdominal wall separate from the umbilicus [11], and amniotic rupture in the pars flaccida part of the umbilical cord secondary to a genetic predisposition or exogenous factors e.g., toxins, drugs, viruses, or radiation [12].

More recently, Lubinsky proposed a binary vascular–thrombotic model for gastroschisis, where the normal involution of the umbilical vein creates a potential site for thrombosis adjacent to the umbilical ring. Subsequent thrombosis, associated with factors increasing maternal estrogen levels, weakens the umbilical ring, creating a site for potential herniation. This model can explain the morphological findings (location at the umbilical ring, typical right-sidedness, and amniocyte inclusions) as well the epidemiologic risk factors (rising incidence due to increasing environmental contamination with estrogen disruptors) [13]. However, evidence for this to be occurring in humans is lacking.

Recent human evidence appears to support the theory that gastroschisis is not a defect of the abdominal wall, but an abnormality of the rudimentary umbilical ring, resulting in a separation of the fetal ectoderm from the amnion’s epithelium on the right side [12,14,15].

## 4. Epidemiology

The epidemiology of gastroschisis has been a subject of considerable interest, given the steady increase in its prevalence over the past three decades. In the US, the prevalence of gastroschisis nearly doubled during the period from 1995 to 2005 [16]. However, this increasing trend is not universal; low and stable rates have been reported from Southern Europe (particularly Italy) and the Middle East [17]. The current prevalence rate of gastroschisis in the US is 4.5 per 10,000 live births [18].

Gastroschisis is unusual among birth defects in that it disproportionately affects younger mothers, with the highest prevalence among mothers aged <20 years (15.7 per 10,000 live births) [17,18,19,20,21,22]. Approximately 70% of infants with gastroschisis are born to women under 25 years of age, and the incidence among teenage mothers is more than seven times that among mothers aged ≥25 years [18]. The prevalence rates are higher among white and Hispanic mothers than among black mothers [17,18,21].

In a recent meta-analysis of 29 studies, maternal smoking (relative risk (RR), 1.56; 95% confidence interval (CI), 1.40–1.74), illicit drug use (RR, 2.14; 95% CI, 1.48–3.07), and alcohol consumption (RR, 1.39; 95% CI, 1.13–1.70) during early pregnancy were associated with an increased risk of gastroschisis [23]. A younger age of the father has been found to be a risk factor in some studies [24,25,26,27]. A recent ecologic analysis found that gastroschisis prevalence was higher in areas with high or medium opioid prescription rates, compared with that in areas with low rates [18]. In a Canadian cohort study involving two large databases, smoking (odds ratio (OR), 2.86; 95% CI, 2.22–3.66), a history of pregestational or gestational diabetes (OR, 2.81; 95% CI, 1.42–5.5), and use of medication to treat depression (OR, 4.4; 95% CI, 1.38–11.8) emerged as significant associations with gastroschisis pregnancies [20]. The National Birth Defects Prevention Study 1997–2011 reported genitourinary tract infections during the periconceptional period to be significantly associated with gastroschisis (OR, 1.5; 95% CI, 1.3–1.9) [28].

There appears to be a spatial variation in the rates of gastroschisis, with “case clusters” in discrete areas relative to surrounding areas [27,28,29,30], suggesting the possible role of agrichemicals as an etiologic factor [31,32].

## 5. Types of Gastroschisis

First proposed by Molik et al. [33], gastroschisis is classified into simple and complex types based on the condition of the bowel. In simple gastroschisis, the bowel is in good condition with no intestinal complications. Complex gastroschisis, on the other hand, is gastroschisis associated with congenital intestinal complications in the form of an atresia, perforation, ischemia, necrosis, or volvulus. Closed or closing gastroschisis is a subset of complex gastroschisis, in which the abdominal wall defect closes around the prolapsed bowel, resulting in exit and/or entry intestinal stricture, atresia, ischemia, necrosis, or resorption. These intestinal complications result from a combination of exposure to digestive compounds in the amniotic fluid and ischemia due to mesenteric constriction at the level of the defect. Infants with closing gastroschisis tend to have a high incidence of short-bowel syndrome (SBS). In extremely-rare cases, known as the “vanishing gut syndrome”, the abdominal wall defect closes completely prenatally, resulting in extreme short gut [34]. In a recent meta-analysis [35], infants with complex gastroschisis, occurring in 17% of cases, have significantly higher mortality rates (16.67%) compared to those with simple gastroschisis (2.18%). Infants with complex gastroschisis also suffer from significantly-higher rates of morbidities. They are started on enteral feeds later, they take longer to reach full enteral feedings, and require a longer duration of parenteral nutrition (PN). They have a higher risk of sepsis, SBS, and necrotizing enterocolitis (NEC). They tend to stay longer in the hospital and are more likely to be sent home with enteral tube feedings and PN. Left-sided gastroschisis is a rare entity, where the abdominal wall defect occurs in the left paraumbilical region. It is more common in females, and associated with a higher incidence of extraintestinal anomalies compared with the right-sided lesions [36].

## 6. Prenatal Diagnosis

Gastroschisis can be diagnosed on prenatal ultrasound scans as early as 12 weeks’ gestation [37]. For a suspected gastroschisis, special attention must be paid to the following features on sonography: (1) absence of a covering membrane or a sac; (2) identification of the site of cord insertion relative to the defect (the defect is paraumbilical, most often right-sided); (3) identification of eviscerated organs; (4) appearance of the eviscerated bowel, e.g., dilation and/or thickening; and (5) identification of associated malformations [38]. Gastroschisis may be associated with gastrointestinal anomalies, such as intestinal atresia, stenosis, or malrotation. Most studies do not include ileal atresia as a separate malformation in infants with gastroschisis; rather it is considered to be a sequence resulting from the primary defect. Approximately 5–15% of patients with gastroschisis have associated extraintestinal anomalies, but recognizable syndromes or chromosomal anomalies are rare [4,39,40,41]. However, in an analysis of a subset of 23 fetuses who experienced an atypical perinatal event (spontaneous abortion, stillbirth, termination of pregnancy, or death within 24 h of birth), the rate of associated malformations was as high as 74%, which may represent a “hidden mortality” that may not be apparent without postmortem examination [41]. A prenatal diagnosis of gastroschisis should therefore prompt a thorough evaluation for associated malformations, including karyotype, ultrasound scan, and fetal echocardiography, in an effort to facilitate appropriate prenatal counselling and decision-making.

Ultrasound in the third trimester is sensitive but has a low positive predictive value and low accuracy for the diagnosis of small for gestational age (SGA) at birth for fetuses with gastroschisis [42].

Maternal serum alpha fetoprotein (MSAFP) levels are usually elevated in abdominal wall defects, including gastroschisis [43].

## 7. Management during Pregnancy

Once the diagnosis of gastroschisis is made, a multidisciplinary team, including the obstetrician, neonatologist, pediatric surgeon and social worker, should provide initial counselling and be involved in the ongoing care of the patient and her fetus. Prenatal ultrasonography may be helpful in identification of reliable predictors of postnatal outcome, and thus facilitate prenatal counselling. A recent meta-analysis of 26 studies involving over 2000 fetuses [44] found significant positive associations between intra-abdominal bowel dilatation and bowel atresia (OR, 5.48; 95% CI, 3.1–9.8), polyhydramnios and bowel atresia (OR, 3.76; 95% CI, 1.7–8.3), and gastric dilatation and neonatal death (OR, 5.58; 95% CI, 1.3–24.1). However, extra-abdominal bowel dilatation is a frequent and inconsequential finding on ultrasound scans [45].

A significant number of newborns with gastroschisis are SGA, with approximately 47 to 61% of the neonates weighing at or below the 10th percentile at birth [46,47,48,49]. In a retrospective study among infants with gastroschisis, being SGA at birth was associated with a four-fold increase in odds for prolonged length of stay (LOS) in the hospital [50].

## 8. Timing and Mode of Delivery

Approximately 30 to 40% of pregnancies with gastroschisis go into spontaneous preterm labor and delivery, compared to 6% in the controls [51,52,53]. The higher rates of preterm labor in these patients have been attributed to the presence of increased levels of pro-inflammatory cytokines (including interleukin-6 and interleukin-8) in the amniotic fluid [54,55]. It has been observed that spontaneous preterm labor is associated with more severely damaged bowel loops, bowel occlusion, and stained amniotic fluid, possibly related to repeated fetal vomiting of the gastrointestinal contents into the amniotic fluid, thus increasing the amount of inflammatory mediators [56]. The incidence of intrauterine fetal death (IUFD) in pregnancies complicated by gastroschisis is approximately 5%, which is significantly higher than in uncomplicated pregnancies [57,58]. The increased stillbirth rate may be related to umbilical cord compression due to acute extra-abdominal bowel dilatation, oligohydramnios, cytokine-mediated inflammation, or volvulus and vascular compromise [57].

The primary determinant of outcome in infants with gastroschisis is the extent of intestinal injury that occurs during fetal life, which is likely due to a combination of exposure of the bowel to amniotic fluid and strangulation of the bowel at the constricting abdominal wall defect. Although a recent systematic review and meta-analysis by Landisch et al. [59] found that elective preterm delivery (<37 weeks) was associated with shorter time to first enteral feed and decreased risk of neonatal sepsis, these benefits were not found in other studies [60,61,62]. Based on the findings of their large retrospective cohort study, Sparks et al. [58] recommended delivery as early as 37 weeks in order to minimize prenatal and postnatal mortality for fetuses with gastroschisis. A decision and cost-effectiveness analysis by Harper et al. [63] found that delivery at 38 weeks was the most cost-effective strategy, with decreased risk of stillbirth and infant death and minimal increase in the number of cases of respiratory distress syndrome (RDS). A large, multicenter randomized controlled trial, the Gastroschisis Outcomes of Delivery (GOOD) study, is ongoing to compare outcomes (IUFD, neonatal death, respiratory morbidity, gastrointestinal or GI morbidity, and sepsis) after delivery at 35 weeks in stable patients with gastroschisis with those following observation and expectant management with a goal of delivery at 38 weeks’ gestation [64].

The optimal mode of delivery of prenatally-diagnosed gastroschisis has been a subject of several observational studies, systematic reviews, and meta-analyses [65,66,67,68,69,70,71]. The mode of delivery was not significantly associated with overall mortality, NEC, secondary repair, sepsis, short gut syndrome, time until full enteral feeding, or LOS. In one study [65], cesarean section was identified as an independent risk factor for the development of respiratory distress at birth (OR, 7.11; 95% CI, 1.06–47.7). Therefore, planned cesarean section in the absence of the usual obstetric indications is not generally recommended.

## 9. Postnatal Management

In the delivery room, it is critical to protect the herniated bowel by covering it in warm, saline-soaked gauze, placing it in a central position on the abdominal wall and covering with a plastic wrap or a plastic bag to decrease evaporative heat and fluid losses. The infant should preferably be positioned in the right lateral decubitus position to prevent vascular damage because of twisting of the mesenteric vascular pedicle.

While it is important to maintain adequate intravascular volume and intestinal perfusion, routine aggressive fluid resuscitation or excessive maintenance fluids should be avoided, and fluid boluses should be used only if there is clinical evidence of hypovolemia and metabolic acidosis. A multivariate analysis of 362 infants with gastroschisis demonstrated a significant, direct relationship between resuscitative fluid volume and increased likelihood of adverse outcomes, including days of post-closure ventilation, total PN or TPN, LOS, and bacteremic episodes [72].

Infants with gastroschisis tend to have elevated C-reactive protein (CRP) values as well as immature to mature neutrophil count (I:T) ratio, but these are not reliable markers of infection or adverse outcomes. Therefore, empiric sepsis evaluation and antibiotic use immediately following delivery may be unnecessary [73,74].

## 10. Surgical Management

The goals of surgical management of gastroschisis include reduction of the herniated viscera into the peritoneal cavity while avoiding direct trauma to the bowel and excessive intra-abdominal pressure, and closure of the abdominal wall defect. While the condition of the exposed bowel and the degree of abdominovisceral disproportion primarily dictate the type and timing of surgical intervention, other factors such as gestational maturity and the infant’s weight and co-morbidities also need to be considered. The surgical options for closure include:(a)Primary reduction, with either immediate sutured closure or sutureless closure;(b)Prosthetic silo placement, gradual visceral reduction followed by delayed sutured or sutureless closure.

### 10.1. Primary Reduction

Primary reduction of the gastroschisis is usually performed if the herniated bowel can be safely placed back into the abdominal cavity without causing excessive intra-abdominal pressure. If the herniated bowel loops appear near-normal, this can be performed in the neonatal intensive care unit (NICU) under mild sedation/analgesia. Although rare, in infants with closing gastroschisis, the narrow fascial defect can compromise intestinal blood flow, and may require urgent enlargement of the defect to preserve bowel perfusion and facilitate reduction.

“Abdominal compartment syndrome”, a serious and potentially-life-threatening complication after primary closure, is characterized by respiratory compromise, and/or lower limb, renal and intestinal ischemia. Intra-gastric or intra-vesical pressures >20 mmHg, or central venous pressure >4 mmHg have been shown to correlate with decreased perfusion to the kidneys and bowel, and potential risk of compartment syndrome [75,76]. Peak inspiratory pressures of <25 cm H_2_O on the ventilator after closure also predicts low risk for abdominal compartment syndrome [77]. Following the reduction of the bowel into the abdominal cavity, the defect is closed either by sutured fascial closure technique or by sutureless closure technique.

### 10.2. Staged Reduction

Staged reduction is achieved by the placement of a spring-loaded silo that provides coverage to the exposed bowel, with subsequent definitive closure of the defect. This method can be used as initial therapy and also following failure of primary reduction. The spring-loaded ring of the silo is inserted through the abdominal wall defect and rests beneath the fascia inside the abdominal cavity without the need for placement of fascial sutures. The procedure can be performed at the bedside under mild sedation/analgesia, without the need for endotracheal intubation. The transparent bag covers the eviscerated bowel, which is reduced daily by tying umbilical tapes around the bag (Figure 2). Once the bowel is completely reduced into the abdomen (Figure 3), closure is performed using either the sutured fascial closure or the sutureless closure techniques.

### 10.3. Sutureless Closure

First described by Sandler et al. [78], the “plastic” sutureless closure technique utilizes the umbilical cord, left deliberately long at the time of birth, as a biologic dressing. After careful reduction of the eviscerated bowel into the abdomen, the gastroschisis defect is covered with the umbilical cord cut and tailored to fit the opening. A clear plastic dressing (Tegaderm 3M) is placed over the defect, which is then allowed to heal by secondary intention. The umbilical-cord-covered defect contracts circumferentially resulting in a scarless abdomen and a cosmetically-acceptable appearance of the umbilicus in 2–4 weeks (Figure 4). This technique can be used after primary reduction as well as following staged reduction with silo placement. In the latter case, the cord is wrapped in Vaseline gauze and kept moist while the silo is in place or placed inside the silo to maintain viability.

### 10.4. Primary Reduction vs. Staged Reduction

The optimal timing of abdominal wall closure in gastroschisis remains debatable, with general agreement that early reduction is best.

Staged reduction with silo placement has the theoretical advantage of achieving reduced intra-abdominal pressure at the time of definitive closure, leading to improved splanchnic perfusion, resulting in faster return of bowel function, reduced rates of infection and NEC and decreased risk of long-term bowel dysfunction [79,80]. Placement of a silo also allows for ongoing assessment of bowel perfusion through the transparent bag. Prolonged use of the silo, however, can lead to pressure necrosis around the silo ring.

In a meta-analysis that included studies with least selection bias, staged closure with silo was associated with better outcomes and a significant reduction in ventilator days, time to first feed, and infection rates [81]. However, in the same meta-analysis, when all studies were included, primary closure was associated with improved outcomes.

A recent large, multicenter retrospective observational study involving 866 neonates with gastroschisis compared infants who underwent immediate closure with those who had a silo placed for ≤5 days [82]. The two groups had generally-equivalent outcomes, except for a higher incidence of ventral hernias in infants who underwent immediate closure compared to those who had silo placed for a short duration. There were no significant differences between the two groups in terms of mortality, time to tolerance of full enteral feeds, duration of TPN, or hospital LOS. Infants who require a silo for longer than 5 days likely represent a subset of patients with more severe disease and less favorable outcomes, and therefore, may not be appropriate to compare to those infants who underwent immediate closure.

Two small prospective randomized trials demonstrated no difference between infants undergoing primary closure and those undergoing silo placement with delayed closure in terms of time to full enteral feeds, duration of ventilatory support, LOS, or incidence of sepsis [83,84]. In one of these studies [84], patients in the silo group were closed at a median of 4 days after silo placement (interquartile range (IQR), 2–5.8 days).

### 10.5. Sutured Closure vs. Sutureless Closure

The sutureless closure method has gained popularity in recent years due to its simplicity, and purported advantages of the ability to perform the procedure at the bedside without the need for general anesthesia, lower intra-abdominal pressures after closure, shorter duration of mechanical ventilation, decreased need for pain medication, superior cosmetic appearance of the umbilicus, and lower hospital cost [78,85,86].

In a large, single-center cohort study of 97 infants with gastroschisis, sutureless repair was associated with significant reduction in duration of mechanical ventilation and pain medication requirements, but with an increased risk of umbilical hernias, compared to sutured closure [87]. In a prospective randomized controlled trial involving 39 infants with gastroschisis, sutureless repair was associated with a significant increase in time to full feeds and time to discharge [88]. A meta-analysis of studies comparing flap closure with fascial closure for gastroschisis by Yousseff et al. demonstrated no significant differences in mortality, LOS, or feeding parameters between the two groups [89]. The infants who underwent flap closure had significantly fewer wound infections, but had an increased risk of umbilical hernia.

In infants with gastroschisis with associated ileal atresia, the surgical management must be individualized based on gestational maturity, birth weight, clinical status, and the condition of the bowel. If the bowel is in good condition, primary anastomosis with abdominal wall closure can be performed. Otherwise, options include creation of stoma with closure or reduction of unrepaired bowel into the abdomen with closure and delayed exploration to establish bowel continuity [77]. The musculature of the abdominal wall is normal in gastroschisis.

Figure 5 provides an algorithm for the choice of optimal surgical procedure based on the type of gastroschisis and the ability to safely reduce the bowel.

## 11. Postoperative Management

The postoperative course in infants with gastroschisis is primarily dependent upon the extent and severity of intestinal injury. Most infants tend to have some degree of adynamic ileus, intestinal dysmotility, and nutrient malabsorption and require nasogastric decompression via a Replogle (nasogastric) tube and TPN via a secure central venous catheter, either a peripherally-inserted central catheter (PICC) or a tunneled Broviac catheter. Feeding is initiated upon return of bowel function (resolution of abdominal distension and bilious gastric drainage). The time to initiation of feeds can range from one to two weeks, or longer, depending on the condition of the bowel. Early initiation of trophic feeds can improve peristalsis, prevent villous atrophy, and reduce bacterial overgrowth. Feeds are advanced slowly as feeding intolerance is common due to intestinal dysmotility. Feeding with maternal expressed breast milk, when available, may help to protect the infant from developing NEC [90]. Early oral stimulation is important to preserve sucking/swallowing reflexes and prevent oral aversion. Infants with gastroschisis have been shown to have abnormal esophageal motor function leading to delayed acquisition of oral feeding milestones, and the possible need for chronic tube feeding [91]. In a multicenter, randomized, double-blind, placebo-controlled trial, enterally-administered erythromycin conferred no advantage in the time taken to achieve full enteral feeding after primary repair of uncomplicated gastroschisis [92].

There appears to a wide variation in the use of prophylactic antibiotics following abdominal wall closure [93]. Based on a multi-institutional review of practice and outcomes, it is recommended that antibiotics be discontinued within 48 h of abdominal wall closure in the absence of culture-positive sepsis or clinical instability [94].

The incidence of NEC in infants with gastroschisis is approximately 5%, which is higher than that predicted based on their gestational age. However, NEC in gastroschisis tends to be clinically less severe than that seen in preterm infants, with only one-quarter of these infants requiring surgery for NEC [95].

Despite infants with gastroschisis having some degree of malrotation, the incidence of midgut volvulus is relatively low at approximately 1% [96]. This may relate to bowel inflammation and handling leading to formation of protective adhesions. Given the low risk of midgut volvulus and the risk of bowel adhesions and obstruction, Ladd’s procedure is not routinely performed in gastroschisis. However, the diagnosis of volvulus must be considered in any infant with gastroschisis presenting with delayed bowel obstruction.

SBS can occur in infants with complex gastroschisis with extensive bowel injury resulting from jejunoileal atresia, midgut volvulus, NEC, or refractory intestinal failure. These infants are dependent on PN for prolonged periods and therefore require placement of indwelling central venous catheters. As a result, they can develop intestinal bacterial overgrowth, bacteremia due to translocation of enteric bacteria, or intestinal failure-associated liver disease. The risk of liver injury can be reduced by using ursodeoxycholic acid, which acts by displacing hydrophobic, more toxic bile salts [97], use of SMOFlipid, an intravenous lipid emulsion containing soybean oil, medium-chain triglycerides, olive oil, and fish oil [98], and cycling of PN to reduce hepatic steatosis [99]. Using enteral antibiotics and probiotics can be helpful in treating bacterial overgrowth. Other pharmacologic interventions used in SBS include use of gastric-acid-suppressing agents, e.g., H_2_-receptor blockers and proton pump inhibitors, prokinetic agents such as erythromycin, anti-diarrheal agents such as loperamide and codeine, absorbent agents such as pectin and guar gum, or cholestyramine [100]. The surgical approaches to SBS include intestinal lengthening procedures, e.g., longitudinal intestinal lengthening and tailoring (LILT) and serial transverse enteroplasty (STEP), and in patients with irreversible hepatic and intestinal failure, intestinal transplantation [101]. Management of SBS often requires multidisciplinary “intestinal failure” teams that include neonatologists, gastroenterologists, surgeons, nutritionists, pharmacists, nurses, and social workers.

Cryptorchidism is common in gastroschisis, occurring in approximately 35% of all male infants [102,103]. It has been postulated that increased intra-abdominal pressure is responsible for the normal descent of the testis, and in gastroschisis, forces that promote testicular descent are absent. However, there is a high rate of spontaneous migration during the first year of life, with nearly 50% of the testes relocating without intervention [102,103]. The watch-and-wait approach is recommended for management of cryptorchidism in gastroschisis due to a high rate of spontaneous migration during the first year of life and greater than 90% of testes being viable on follow-up. Laparoscopic orchidopexy is a safe and feasible option for testes that remain intraabdominal at follow-up [103]. In contrast, the rate of spontaneous testicular descent was only 10% in boys with omphalocele [102].

## 12. Prognosis and Outcome Prediction

The prognosis of infants with gastroschisis is largely dependent on the condition of the bowel at birth. In patients with simple gastroschisis, the mean LOS is 41 ± 32 days and the mortality rate is 3.4%, while patients with complex gastroschisis have a mean LOS of 85 ± 60 days and a mortality rate of 9.3% [104]. Approximately 25% of infants with simple gastroschisis and over 70% of infants with complex gastroschisis will develop subsequent bowel obstruction resulting from adhesions, anastomotic stricture, or volvulus, requiring repeat surgical interventions [105]. In a retrospective study of 83 infants with complex gastroschisis, about one-third of these infants required home PN, and two-thirds were discharged on gastrostomy feeds [106]. The Gastroschisis Prognostic Score (GPS), originally developed by Cowan et al. [107], is a validated visual bowel-injury-scoring tool performed at the bedside shortly after birth that assesses the presence and severity of bowel necrosis, matting, atresia, and perforation (Table 1). In a study involving 849 survivors of gastroschisis, for each integer increase in GPS score (i.e., increase in GPS score by 1), there was an increase in the LOS by 17 days, and an increase in the duration of PN by 13 days over baseline values in low-risk infants (with GPS of 0 or 1) who have LOS of one month and need 3 weeks of PN [108].

Intestinal fatty-acid-binding protein (I-FABP) is present in the epithelial cells of the intestine, and is released into the circulation after enterocyte damage. Urinary I-FABP has been shown to be a marker for intestinal mucosal damage, with significantly-higher levels in infants with complex gastroschisis compared to those with simple gastroschisis. However, urinary I-FABP was not useful for outcome prediction; it failed to predict early start of minimal enteric feeding, full enteral feeding, or LOS [109].

## 13. Medium- and Long-Term Outcomes

Advances in the management of gastroschisis have led to a marked improvement in survival rates, which are well over 90% in most centers in North America [104]. However, infants with gastroschisis are at risk of suboptimal growth and adverse neurodevelopmental outcomes due to multiple factors, including prematurity, SGA, need for prolonged PN, recurrent infections, and prolonged hospital stays. There are also concerns about the effects of prolonged or repeated exposures of anesthetic agents on the developing brain, especially in infants who require multiple surgical interventions [110,111]. The follow-up studies on gastroschisis are, unfortunately, sparse, with small sample sizes and many with suboptimal follow-up [112,113,114,115,116,117,118,119,120].

While a large proportion of infants with gastroschisis exhibit suboptimal weight gain during the first 2 years [112,113], most will experience “catch-up” growth, with significant improvements in z-scores and growth percentiles on continued follow-up [114,115,116]. In a follow-up study of 50 children born with gastroschisis, Harris et al. found only 2% were underweight on follow-up at a median age of 9 years. However, children with complex gastroschisis had a significantly-lower weight z scores compared to those with simple gastroschisis [115]. Davies et al. interviewed 23 children born with gastroschisis at a median age of 16 years and found 96% were in good health with growth within normal limits [116].

The neurodevelopmental outcomes of patients with gastroschisis at 1–2 years of follow-up are similar to those of non-surgical neonatal intensive care unit graduates of similar birth weight and gestational maturity [112,113,114,117,118]. A cohort study of 42 infants with gastroschisis by Harris et al. showed normal IQ scores at a median age of 10 years; however, there was a small but significant decrease in the working memory index, as well as impairments in behavior and parent–child relationship [119]. Lap et al. found that school-aged children born with gastroschisis had significantly-lower scores on several aspects of attention, response inhibition, executive functioning, verbal intelligence, and fine motor skills compared to matched controls [120]. van Manen et al. reported visual impairment in 5% and sensorineural hearing loss in 10% of the survivors at a median age of 20 months [114]. As would be expected, children with complex gastroschisis tend to have worse neurodevelopmental outcomes compared to those with simple gastroschisis [121]. Despite worse short-term outcomes and the need for more surgical interventions among patients with complex gastroschisis, the parent-reported quality of life scores were similar to those with simple gastroschisis [122,123].

In an observational, longitudinal cohort study, Snoep et al. found that adolescents and adults with a history of gastroschisis have health-related quality of life (HRQoL) comparable to healthy controls [124]. At a median age of 9 years, Harris et al. found that 41% of children with history of gastroschisis were suffering from weekly bouts of abdominal pain [115]. Recurrent abdominal complaints (>once a week) during the teenage years and beyond included abdominal pain (14%), gas bloat (36%), difficulty with completing a meal (21%), and gastroesophageal reflux symptoms (43%) [124].

## 14. Summary

Gastroschisis is a congenital anomaly in the rudimentary umbilical ring, most often just to the right of a normally-inserted umbilical cord. There is herniation of a variable amount of viscera through this defect, with no covering membrane or sac. The prevalence of gastroschisis has been steadily increasing, presumably due to environmental factors. Based on the absence or presence of intestinal complications, gastroschisis is classified into simple and complex types. Prenatal ultrasound scans can diagnose gastroschisis as early as 12 weeks of gestation. Routine preterm delivery or elective cesarean section have not been shown to improve outcomes. Postnatal surgical management is directed towards reduction of the herniated viscera and closure of the abdominal wall. While the overall survival rates are well over 90%, prognosis depends on the condition of the bowel at birth. Infants with significant bowel damage at birth are “at-risk” for early death or adverse long-term outcomes.

## Figures and Tables

**Figure 1 children-07-00302-f001:**
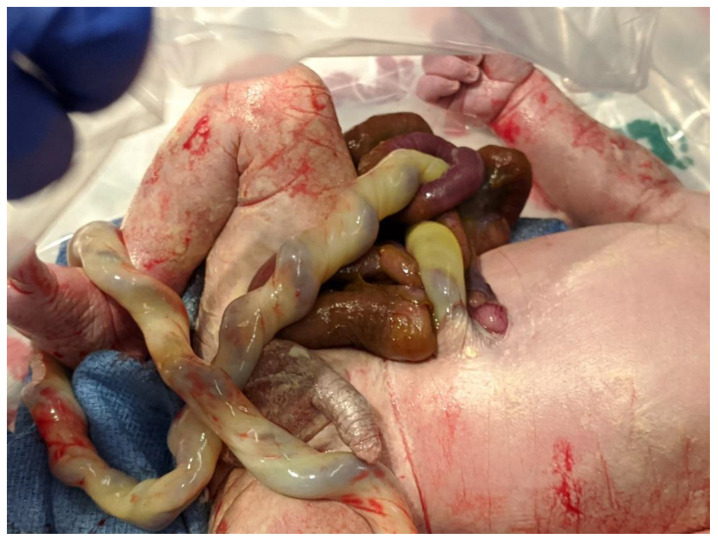
Term infant with simple gastroschisis. The picture shows a small abdominal wall defect to the right of the site of umbilical cord insertion.

**Figure 2 children-07-00302-f002:**
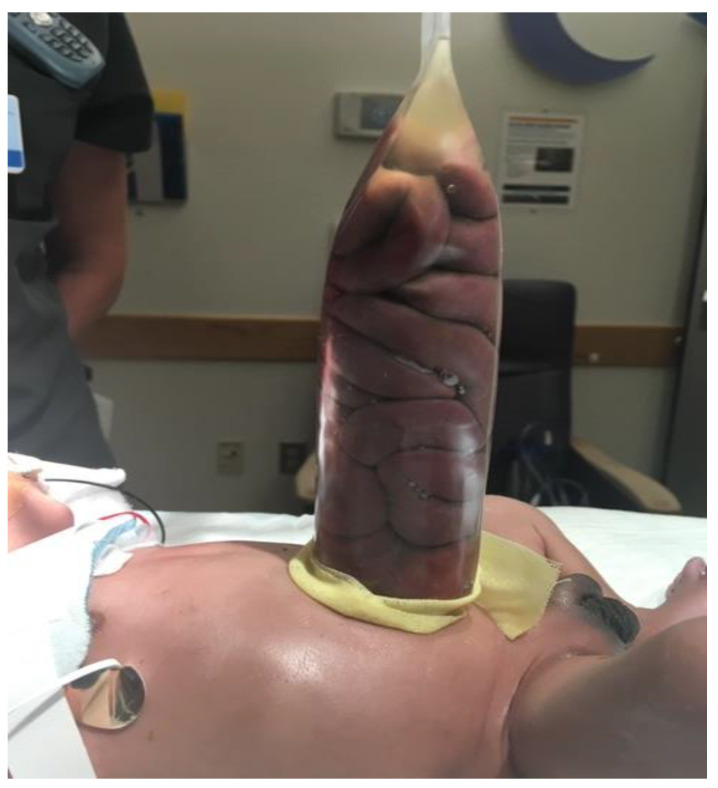
Staged reduction of gastroschisis. Shown is a picture with the bowel loops placed in a silo.

**Figure 3 children-07-00302-f003:**
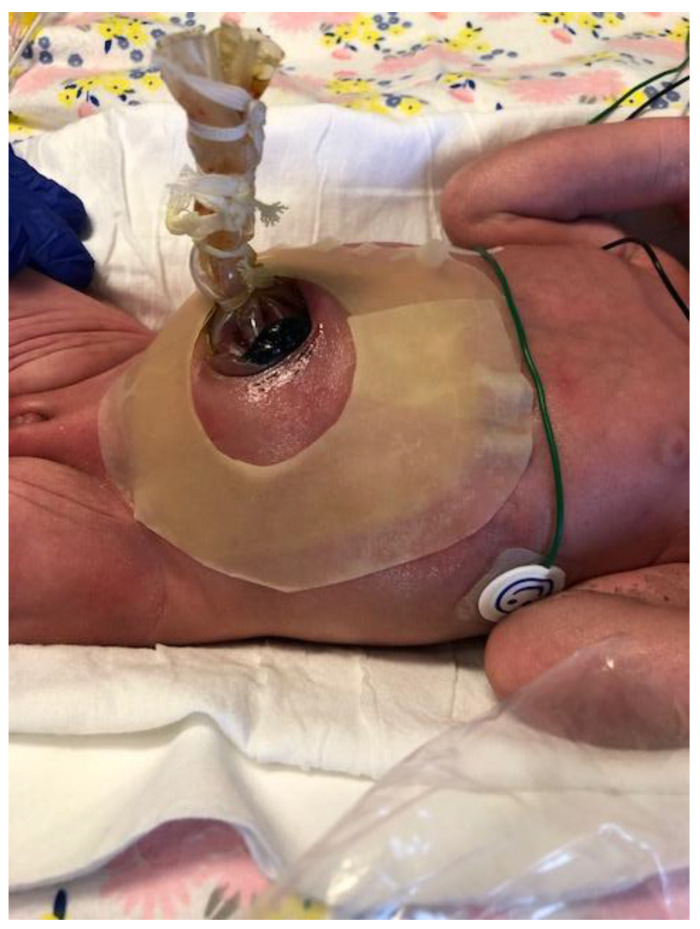
Staged reduction of gastroschisis. Shown is a picture with the bowel loops reduced into the abdomen using a silo.

**Figure 4 children-07-00302-f004:**
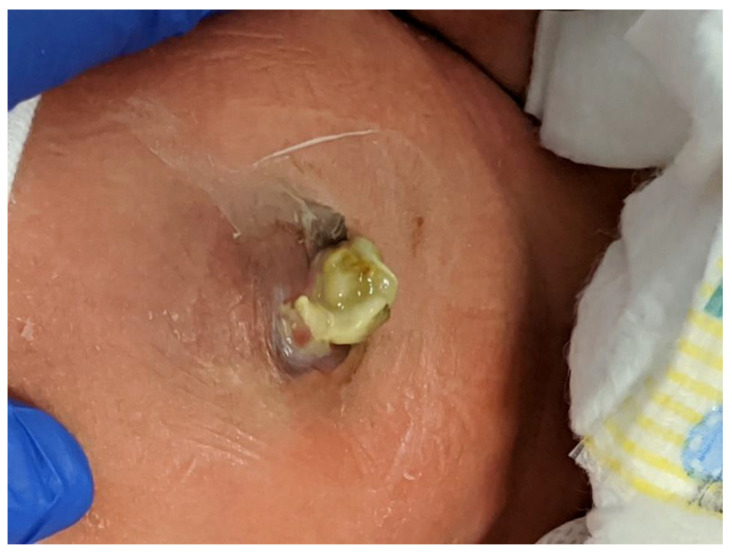
The infant shown in Figure 1 underwent a primary sutureless closure soon after birth. Shown at 19 days of age, he is on full oral feeds.

**Figure 5 children-07-00302-f005:**
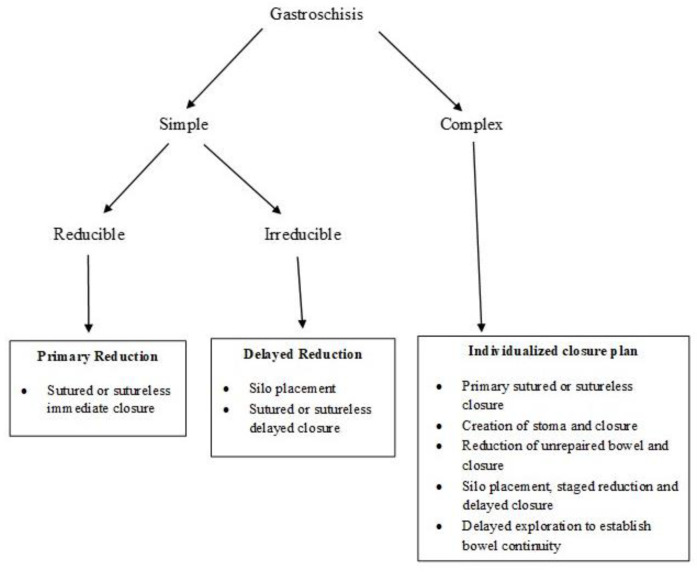
Algorithm for reduction and closure of gastroschisis. (Adapted from Ref. [77]).

**Table 1 children-07-00302-t001:** The Gastroschisis Prognostic Score (GPS).

Parameter	Description	Score	Description	Score	Description	Score
*Matting*	None	0	Mild	1	Severe	4
*Atresia*	None	0	Suspected	1	Present	2
*Perforation*	None	0			Present	2
*Necrosis*	None	0			Present	4

Adapted from Ref. [108].

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
