# Peer review of "Gastroschisis: A State-of-the-Art Review"

_children, 2020, doi:10.3390/children7120302_

Round 1

Reviewer 1 Report

Great review, well written and very comprehensive.

Minor Comment:

In prenatal diagnosis part please add:

Ultrasound in the third trimester is sensitive but had a low positive predictive value and low accuracy for the diagnosis of SGA at birth for fetuses with gastroschisis.

Accuracy of Ultrasound to Predict Neonatal Birth Weight Among Fetuses With Gastroschisis: Impact on Timing of Delivery.

Fisher JE, Tolcher MC, Shamshirsaz AA, Espinoza J, Sanz Cortes M, Donepudi R, Belfort MA, Nassr AA.

J Ultrasound Med. 2020 Oct 1. doi: 10.1002/jum.15519. Online ahead of print.

PMID: 33002208

Author Response

Reviewer # 1

C1: Great review, well written and very comprehensive.

R1: Thank you for your positive comments on our manuscript.

C2: Minor Comment:

In prenatal diagnosis part please add:

Ultrasound in the third trimester is sensitive but had a low positive predictive value and low accuracy for the diagnosis of SGA at birth for fetuses with gastroschisis.

Accuracy of Ultrasound to Predict Neonatal Birth Weight Among Fetuses With Gastroschisis: Impact on Timing of Delivery.

Fisher JE, Tolcher MC, Shamshirsaz AA, Espinoza J, Sanz Cortes M, Donepudi R, Belfort MA, Nassr AA.

J Ultrasound Med. 2020 Oct 1. doi: 10.1002/jum.15519. Online ahead of print.

PMID: 33002208

R2: As suggested, we have added the reference in the section of “Prenatal Diagnosis” in the revised manuscript.

Reviewer 2 Report

The manuscript is well-written and structured with up-to-date references, and I only miss some few aspects.

In cases with a very narrow that may compromise intestinal flow, an immediate widening of the opening with a scissor may be life-saving – especially when the baby has to be transported to another hospital to final surgery.

What about antibiotics both pre- and postoperatively?

How to deal with the intra-abdominal placed testis that occur in some kids?

In patients with intestinal atresia would you recommend primary anastomosis or stoma. Should both a proximal and distal stoma bee placed? Or would you recommend primary abdominal closure and delayed anastomosis?

Author Response

Reviewer # 2

C1: The manuscript is well-written and structured with up-to-date references, and I only miss some few aspects. 

R1: Thank you for your positive comments on our manuscript.

C2: In cases with a very narrow that may compromise intestinal flow, an immediate widening of the opening with a scissor may be life-saving – especially when the baby has to be transported to another hospital to final surgery. 

R2: As suggested, this has been added under “Primary Reduction” in the revised manuscript.

C3: What about antibiotics both pre- and postoperatively? 

R3: Preoperative antibiotics has now been included in Section 9. Postnatal management and Postoperative antibiotics has now been included in Section 11. Postoperative management, in the revised manuscript.

C4: How to deal with the intra-abdominal placed testis that occur in some kids? 

R4: This has been discussed in Section 11. Postoperative management, in the revised manuscript.

C5: In patients with intestinal atresia would you recommend primary anastomosis or stoma. Should both a proximal and distal stoma be placed? Or would you recommend primary abdominal closure and delayed anastomosis?

R5: In infants with gastroschisis with associated ileal atresia, the surgical management must be individualized based on gestational maturity, birth weight, clinical status and the condition of the bowel. If the bowel is in good condition, primary anastomosis with abdominal wall closure can be performed. Otherwise, options include creation of stoma with closure or reduction of unrepaired bowel into the abdomen with closure and delayed exploration to establish bowel continuity. This has been added to the revised manuscript, in Section 10. Surgical management.

Reviewer 3 Report

The authors provide a review of the pathogenesis, epidemiology of risk factors, definition and types of gastroschisis, prenatal diagnosis and management, delivery, and postnatal care. If state of the art, as stated in the title, the authors should do a more exhaustive review, especially for the pathogenesis, epidemiologic risk factors, other unrelated birth defects. The other sections in this manuscript are certainly more exhaustive.

Abstract

The word “incidence” should be replaced with prevalence since we never know the true incidence of any birth defect. To do so would require the identification and monitoring of all conceptions.

The prevalence of gastroschisis has been increasing for more than 2 decades. Sweden was the first to report an increase decades ago, followed by other countries.

We do not yet understand how genetics may play a role in the etiology of gastroschisis. But, the underlying genetic response to an environmental exposure may be important. I would suggest the authors revise the 3rd sentence. We do know that there are several maternal risk factors associated with increasing the risk of gastroschisis. How genetics may contribute to the risk is unknown, but possible.

The authors do not review the all studies that present data on unrelated birth defects (only one study is cited). What is rare?

Section 2: the International Clearinghouse for Birth Defects Surveillance and Research (ICBDSR). Clearinghouse is one word and Surveillance precedes Research.

Section 3:  The major challenge with gastroschisis is the different hypotheses proposed over several decades on pathogenesis which tends to drive studied risk factors. No doubt, this has been very confusing. Pathogenesis must be considered in light of the normal human embryology. Since there are many papers on this topic, most not based on human evidence, this is a challenging topic. Recent human evidence by Rittler et al (2013), Bargy and Beaudoin (2014), and Beudoin (2017) provide the human evidence that gastroschisis is an abnormality of the rudimentary umbilical ring – resulting in a separation of the fetal ectoderm from the amnion’s epithelium on the right side.

The authors reference Lubinsky (2014) who has not provided any human evidence for his vascular-thrombotic model. The recent human observations (Rittler et al; Bargy and Beaudoin) demonstrates gastroschisis is not a defect of the abdominal wall, nor a disruption of the embryonic vessels. From an embryologic standpoint, the omphalomesenteric vessels and the umbilical vessels do not supply the abdominal wall or the area of the umbilical ring. If the either of these vessels were to be disrupted or clotted, the embryo would not survive since these vessels supply the midgut (omphalomesenteric artery) or drain into the vascular system (umbilical and omphalomesenteric vein fuse to build the portal and hepatic venous system). The abdomen is supplied by the intersegmental arteries that originate from the dorsal aorta. The authors are certainly entitled to their opinion, but this should be supported by what we know about normal human embryology.

Section 4:  The most consistent maternal risk factors are young maternal age, smoking and low/normal BMI and infections seems to be important (but more difficult to evaluate using current techniques - e.g., questionnaire). Many other exposures, such as alcohol, illicit drugs, medications have not been consistently observed to increase risk, some studies finding no increased risk. An exhaustive (state of the art) review of the risk factors is challenging, it might be best for the authors to delete this section of the paper as it is not complete. A review of the epidemiologic evidence would take an entire article focusing only on risk factors.

Section 6:  The authors do not provide an exhaustive review of studies that present data on unrelated birth defects. Some studies suggest ~80% of babies with gastroschisis are truly isolated. The authors only reference one study by Mastroiacova et al (2007) which stated ~85% were isolated. This study was based only on ICD9/BPA coding.

Section 7 or 8 or 11: It may be helpful to address congenital atresia vs. acquired atresia postnatally. Many studies do not distinguish between these but it is important to do so. Researchers will sometimes consider atresia to be a separate defect but, if congenital should be considered as a sequence (resulting from the primary defect, gastroschisis).

Section 10.1: Primary reduction – 2nd paragraph: “serous” should be “serious” I believe

The authors make no mention or review of the abdominal musculature – it would be useful to describe this, as either normal or abnormal.  My understanding from the literature is that the muscles are normal in babies with gastroschisis.

Section 13: The authors state that infants “…are at risk for suboptimal growth and adverse neurodevelopmental outcomes due to multiple factors, including exposures to toxins/drugs in utero…”   Parents of children with gastroschisis have a visceral reaction to these types of comments that are without any evidence… such as the ecologic study on opioids (Short et al, 2019). Unless the authors can reference this statement with a well-designed study that demonstrates prenatal exposures are associated with suboptimal growth and adverse neurodevelopmental outcomes, I would suggest deleting this statement.

Section 14:  Based on the work of both Rittler et al (2013), Bargy and Beaudoin (2014) and Beaudoin (2017), gastroschisis is not a “full-thickness” defect. The extruded bowel is within the umbilical ring and to the right of the umbilical cord, which is appropriately attached on the left side of the umbilical ring.

The word “incidence” should be replaced with prevalence since we never know the true incidence of any birth defect.

Author Response

Reviewer # 3

C1: The authors provide a review of the pathogenesis, epidemiology of risk factors, definition and types of gastroschisis, prenatal diagnosis and management, delivery, and postnatal care. If state of the art, as stated in the title, the authors should do a more exhaustive review, especially for the pathogenesis, epidemiologic risk factors, other unrelated birth defects. The other sections in this manuscript are certainly more exhaustive.

R1: Thank you for your comments. Our initial version of the manuscript was much longer and provided additional details about the aspects of gastroschisis mentioned by the reviewer. However, it was suggested that the manuscript be more a “review” and not a “book chapter”, and hence, it was considerably shortened.

Abstract.

C2: The word “incidence” should be replaced with prevalence since we never know the true incidence of any birth defect. To do so would require the identification and monitoring of all conceptions.

R2: Thank you for your suggestion. This has been done in the revised manuscript.

C3: The prevalence of gastroschisis has been increasing for more than 2 decades. Sweden was the first to report an increase decades ago, followed by other countries.

R3: Thank you for the correction. We have now used the term ‘several’, instead of ‘two’, decades, in the revised manuscript.

C4: We do not yet understand how genetics may play a role in the etiology of gastroschisis. But, the underlying genetic response to an environmental exposure may be important. I would suggest the authors revise the 3rd sentence. We do know that there are several maternal risk factors associated with increasing the risk of gastroschisis. How genetics may contribute to the risk is unknown, but possible.

R4: Thank you for the suggestion. We have revised the sentence to state: “However, the contribution of genetic factors to the overall risk remains unknown.” in the revised manuscript.

C5: The authors do not review the all studies that present data on unrelated birth defects (only one study is cited). What is rare?

R5: We have updated Section 6 by adding and discussing 2 recent studies about extraintestinal anomalies in the revised manuscript.

C6: Section 2: the International Clearinghouse for Birth Defects Surveillance and Research (ICBDSR). Clearinghouse is one word and Surveillance precedes Research.

R6: This has been corrected in the revised manuscript.

C7: Section 3:  The major challenge with gastroschisis is the different hypotheses proposed over several decades on pathogenesis which tends to drive studied risk factors. No doubt, this has been very confusing. Pathogenesis must be considered in light of the normal human embryology. Since there are many papers on this topic, most not based on human evidence, this is a challenging topic. Recent human evidence by Rittler et al (2013), Bargy and Beaudoin (2014), and Beudoin (2017) provide the human evidence that gastroschisis is an abnormality of the rudimentary umbilical ring – resulting in a separation of the fetal ectoderm from the amnion’s epithelium on the right side.

The authors reference Lubinsky (2014) who has not provided any human evidence for his vascular-thrombotic model. The recent human observations (Rittler et al; Bargy and Beaudoin) demonstrates gastroschisis is not a defect of the abdominal wall, nor a disruption of the embryonic vessels. From an embryologic standpoint, the omphalomesenteric vessels and the umbilical vessels do not supply the abdominal wall or the area of the umbilical ring. If the either of these vessels were to be disrupted or clotted, the embryo would not survive since these vessels supply the midgut (omphalomesenteric artery) or drain into the vascular system (umbilical and omphalomesenteric vein fuse to build the portal and hepatic venous system). The abdomen is supplied by the intersegmental arteries that originate from the dorsal aorta. The authors are certainly entitled to their opinion, but this should be supported by what we know about normal human embryology.

R7: We have revised section 3 of the manuscript to reflect the above information provided by the reviewer.

C8: Section 4:  The most consistent maternal risk factors are young maternal age, smoking and low/normal BMI and infections seems to be important (but more difficult to evaluate using current techniques - e.g., questionnaire). Many other exposures, such as alcohol, illicit drugs, medications have not been consistently observed to increase risk, some studies finding no increased risk. An exhaustive (state of the art) review of the risk factors is challenging, it might be best for the authors to delete this section of the paper as it is not complete. A review of the epidemiologic evidence would take an entire article focusing only on risk factors.

R8: We certainly appreciate the fact that a full manuscript can be written discussing the ‘Epidemiology of gastroschisis”. However, we believe that a state-of-the-art review article on this topic does require inclusion, and some discussion, of this aspect of gastroschisis. As the reviewer will note, we have provided information from multiple sources (references and meta-analyses) and provide statistical information about the robustness/validity of the data. Hence, we would suggest this section be retained. If the reviewer/editors do not concur, we are happy to remove the entire section from the manuscript.

C9: Section 6:  The authors do not provide an exhaustive review of studies that present data on unrelated birth defects. Some studies suggest ~80% of babies with gastroschisis are truly isolated. The authors only reference one study by Mastroiacova et al (2007) which stated ~85% were isolated. This study was based only on ICD9/BPA coding.

R9: We have updated this section (Section 6) by adding and discussing 2 recent studies about extraintestinal anomalies in the revised manuscript.

C10: Section 7 or 8 or 11: It may be helpful to address congenital atresia vs. acquired atresia postnatally. Many studies do not distinguish between these but it is important to do so. Researchers will sometimes consider atresia to be a separate defect but, if congenital should be considered as a sequence (resulting from the primary defect, gastroschisis).

R10: Thank you for the suggestion. It has been incorporated in Section 6 of the revised manuscript.

C11: Section 10.1: Primary reduction – 2nd paragraph: “serous” should be “serious” I believe

R11: Thank you for detecting the typographical error. This has been corrected the revised manuscript.

C12: The authors make no mention or review of the abdominal musculature – it would be useful to describe this, as either normal or abnormal.  My understanding from the literature is that the muscles are normal in babies with gastroschisis.

R13: The reviewer is correct. This has now been mention in Section 10.5 of the revised manuscript.

C14: Section 13: The authors state that infants “…are at risk for suboptimal growth and adverse neurodevelopmental outcomes due to multiple factors, including exposures to toxins/drugs in utero…”   Parents of children with gastroschisis have a visceral reaction to these types of comments that are without any evidence… such as the ecologic study on opioids (Short et al, 2019). Unless the authors can reference this statement with a well-designed study that demonstrates prenatal exposures are associated with suboptimal growth and adverse neurodevelopmental outcomes, I would suggest deleting this statement.

R14: We have deleted “exposures to toxins/drugs in utero” from the revised manuscript.

C15: Section 14:  Based on the work of both Rittler et al (2013), Bargy and Beaudoin (2014) and Beaudoin (2017), gastroschisis is not a “full-thickness” defect. The extruded bowel is within the umbilical ring and to the right of the umbilical cord, which is appropriately attached on the left side of the umbilical ring.

R15: This has been corrected the revised manuscript.

C16: The word “incidence” should be replaced with prevalence since we never know the true incidence of any birth defect.

R16: Thank you for your suggestion. This has been done in the revised manuscript.

Round 2

Reviewer 3 Report

The authors have responded to my edits and suggestions. I have no further comments.